A lightweight segmentation network for endoscopic surgical instruments based on edge refinement and efficient self-attention

Zhou Mengyu 1 2
http://orcid.org/0000-0002-1946-2067 Han Xiaoxiang 2
Liu Zhoujin 1
Chen Yitong 1
Sun Liping 1 3 sunlp@sumhs.edu.cn
1 School of Medical Instruments, Shanghai University of Medicine & Health Sciences , Shanghai , P.R.China
2 School of Health Science and Engineering, University of Shanghai for Science and Technology , Shanghai , China
3 School of Information Science and Technology, Fudan University , Shanghai , China
Wan Shibiao
Electronic publication date: 2023 Dec 11
Publication date: 2023
Volume: 9
Electronic Location ID: e1746
Received 2023 Jul 5; Accepted 2023 Nov 17
Copyright: © 2023 Zhou et al.
Copyright year: 2023
Copyright holder: Zhou et al.
License: This is an open access article distributed under the terms of the Creative Commons Attribution License, which permits unrestricted use, distribution, reproduction and adaptation in any medium and for any purpose provided that it is properly attributed. For attribution, the original author(s), title, publication source (PeerJ Computer Science) and either DOI or URL of the article must be cited.
License URL: https://creativecommons.org/licenses/by/4.0/

Keywords: Surgical instruments, Semantic segmentation, Lightweight network, Efficient self-attention

Funding: National Key R&D Program 2018YFB1307700 Shanghai University of Medicine & Health Sciences The National Key R&D Program This work was supported by the National Key R&D Program project (2018YFB1307700). The Excellent Doctoral/Master’s Dissertation Cultivation Project of Shanghai University of Medicine & Health Sciences supported the APC for this article. The National Key R&D Program project was involved in collecting and analyzing the private dataset, as well as acquiring and analyzing experimental data. The funders had no role in study design, data collection and analysis, decision to publish, or preparation of the manuscript.

==============================
In robot-assisted surgical systems, surgical instrument segmentation is a critical task that provides important information for surgeons to make informed decisions and ensure surgical safety. However, current mainstream models often lack precise segmentation edges and suffer from an excess of parameters, rendering their deployment challenging. To address these issues, this article proposes a lightweight semantic segmentation model based on edge refinement and efficient self-attention. The proposed model utilizes a lightweight densely connected network for feature extraction, which is able to extract high-quality semantic information with fewer parameters. The decoder combines a feature pyramid module with an efficient criss-cross self-attention module. This fusion integrates multi-scale data, strengthens focus on surgical instrument details, and enhances edge segmentation accuracy. To train and evaluate the proposed model, the authors developed a private dataset of endoscopic surgical instruments. It containing 1,406 images for training, 469 images for validation and 469 images for testing. The proposed model performs well on this dataset with only 466 K parameters, achieving a mean Intersection over Union (mIoU) of 97.11%. In addition, the model was trained on public datasets Kvasir-instrument and Endovis2017. Excellent results of 93.24% and 95.83% were achieved on the indicator mIoU, respectively. The superiority and effectiveness of the method are proved. Experimental results show that the proposed model has lower parameters and higher accuracy than other state-of-the-art models. The proposed model thus lays the foundation for further research in the field of surgical instrument segmentation.

Introduction

In the past two decades, with the rapid development of robotic technology, surgical robots have brought revolutionary changes to minimally invasive surgery. The da Vinci Surgical System is an outstanding representative, serving as an advanced robotic platform. It not only possesses exceptional flexibility and stability but also offers advantages such as minimal patient trauma, rapid recovery, thorough resection, and reduced complications. Currently, it is widely applied in various subspecialties, including general surgery, obstetrics and gynecology, and cardiac surgery, for both adults and children (Tan & Wang, 2013). To ensure the stable operation of surgical robots, computer-assisted systems have become an indispensable component. Among them, computer vision technology (including object detection and image segmentation) plays a crucial role in tracking and pose estimation of medical instruments. The precise segmentation of surgical instruments is a vital step in computer-assisted systems. This step not only provides surgeons with accurate positional information of surgical instruments but also offers intuitive cues for safe surgical operations. Simultaneously, it evaluates the interaction between surgical instruments and the background tissue during the surgical process (Bouget et al., 2017; Yu et al., 2020).

In specific extreme scenarios, due to lighting conditions and intricate textures, it becomes challenging for the human eye to distinguish between surgical instruments and tissue background. Such circumstances may lead surgical instruments to pull excessively or inadvertently come into contact with human tissues, thereby compromising the overall surgical safety. The objective of surgical instrument segmentation is to accurately distinguish the instrument from the surrounding tissue and present it in a visually clear manner to the surgeon. Surgeons can ascertain the exact location and orientation of surgical instruments. This ensures the safety of the surgery and aids physicians in decision-making. Precise segmentation of surgical instrument images is a pivotal step towards realizing automated robotic surgery, contributing to increased surgical efficiency and accuracy. However, semantic segmentation of surgical instruments poses unique challenges compared to other segmentation tasks in the medical field. Firstly, the distal end of the surgical instrument holds the utmost importance as it enables precise and delicate operations. The functional components crucial to surgical instruments are typically located at the operational end, comprising elements like blades or clamps. In the context of surgical procedures, achieving precise manipulation necessitates the accurate segmentation of these pivotal elements. Edge information serves to delineate the precise outlines of these key instrumental components, thereby enhancing surgical precision. The precise detection of surgical instrument edges facilitates the effective identification and segmentation of the operational end, empowering the model to adeptly adjust to intricate maneuvers within the surgical environment. Consequently, it is designed to be compact and intricately structured. Secondly, the instrument undergoes continuous movement during surgery, resulting in constantly changing orientations.

Surgical instrument segmentation methods can be divided into two categories: traditional image processing-based methods and deep learning-based methods. Traditional methods rely on specific image features such as grayscale and color (Sevak et al., 2017). However, these methods are complicated to process and require manual intervention from professionals with a certain knowledge background. As a result, the segmentation accuracy of these methods is low. On the other hand, deep learning, which is a promising direction in artificial intelligence, has been widely adopted in medical imaging diagnosis (Esteva et al., 2021). Deep learning-based image segmentation methods, when combined with high-performance hardware, can provide an end-to-end detection solution. These methods typically utilize convolutional neural networks, which are capable of effectively processing the original image and extracting meaningful features. Consequently, deep learning-based methods often outperform traditional methods in terms of performance and detection efficiency. In the field of surgical instruments, researchers have proposed various segmentation methods based on deep learning. For instance, Ni et al. (2020) introduced a pyramid attention aggregation network that incorporates a dual attention module and a pyramid upsampling module to capture joint semantic information and global context for modeling. Yang et al. (2022) enhanced a U-net variant network with non-local attention blocks and double attention modules to emphasize the areas containing surgical instruments. Yu et al. (2020) proposed a U-shaped variant network that utilizes dense upsampling convolution instead of deconvolution for sampling, and applies a side loss function to each side output layer.

The current network model used for surgical instruments in surgery has a large number of parameters (“parameters” refers to the variables obtained during the model training process), resulting in a long reasoning time. This poses challenges for deployment in a clinical engineering environment. In our experimental investigation, certain constraints in the existing methodology pertaining to surgical instrument segmentation were noted. These were manifested as inadequacies in the level of detail discernible in the segmentation outcomes. This phenomenon is particularly noticeable in the blurred delineation at the segmented edges or the inability to capture the nuanced structures of the instruments. To address these issues, we propose a lightweight segmentation network for endoscopic surgical instruments. Our network is based on self-attention and efficient edge refinement techniques. In this network, we utilize the lightweight encoder LDCNet, which we previously developed for remote sensing applications (Han et al., 2023). Remote sensing images frequently encompass intricate data, necessitating a proficient encoder for extraction. The architecture of LDCNet leverages the dense connection paradigm of DenseNet (Huang et al., 2017), fostering an enhanced information flow within the network and facilitating the extraction of both global and local features. Remote sensing applications commonly necessitate models to excel in both computational and memory efficiency. LDCNet achieves efficient parameter sharing and reuse by deriving inspiration from the branch topology of ResNeXt (Xie et al., 2017), wherein branch topology denotes parallel connections or multipath designs in neural network structures. This approach substantially diminishes the parameter count of the model. The design principles of the LDCNet encoder harmonize with the requisites of endoscopic image processing; consequently, we persist in employing this encoder. To enhance the model’s focus on surgical instruments and improve segmentation accuracy for instruments with varying shapes and sizes, we introduce the Feature Pyramid Network (FPN) (Lin et al., 2017a) and the Criss-Cross Attention module (CCAM) (Huang et al., 2019). Experimental results demonstrate the effectiveness of our proposed method, which achieves competitive results on both public datasets (Kvasir-instrument, Endovis2017) and private datasets.

The main contributions of this article are as follows: 1. We propose a surgical instrument segmentation algorithm for endoscopic images based on marginalization and efficient self-attention mechanism. Our method employs an encoder-decoder structure, where a lightweight LDCNet acts as an encoder to extract high-quality semantic information with fewer parameters.

2. To enhance the ability of edge segmentation, we propose a novel approach that combines the feature pyramid and an efficient cross-attention module. This allows the model to extract contextual information and focus more on the fine details of surgical instruments’ edges.

3. We verify the effectiveness and feasibility of our proposed method on a private dataset and two public datasets. Compared with several mainstream lightweight semantic segmentation networks, our proposed method accurately segments surgical instruments using fewer model parameters.

Related works

Lightweight networks

In order to ensure easy deployment in clinical engineering, it is crucial to develop a lightweight and efficient model. The ShuffleNet series, known for its effectiveness, offers a solution. ShuffleNetV1 (Zhang et al., 2018) introduced the channel shuffle operation, enabling faster processing through group convolution. Another notable family of lightweight networks is the MobileNet series. MobileNetV1 (Howard et al., 2017) utilizes depth separable convolution to construct a lightweight network. MobileNetV2 (Sandler et al., 2018) introduces the innovative inverted residual with a linear bottleneck unit, improving overall accuracy and speed despite increased layer count. MobileNetV3 (Howard et al., 2019) combines AutoML technology and manual fine-tuning to create an even lighter network. Additionally, EffectNet (Tan & Le, 2019) achieves a balanced approach by uniformly scaling depth, width, and resolution using fixed scaling factors. Lightweight networks, due to their fewer parameters and computational requirements, are suitable for scenarios demanding real-time processing. In the task of surgical instrument segmentation, real-time capabilities and lightness are crucial. These networks have achieved commendable results in terms of speed and parameter quantity, making them beneficial for deployment in surgical robots. However, in the domain of surgical instrument segmentation, the precision of edge segmentation still requires further improvement.

Multi-scale contextual feature extraction

Multi-scale object detection is a critical problem in computer vision. To address this issue, (He et al., 2015) proposed SPPNet, which applied the concept of scale pyramid and feature fusion to convolutional neural networks. The spatial pyramid pooling layer, at the heart of SPPNet, generates consistent-sized vectors for input images of diverse dimensions. This permits the capture of spatial feature details across various scales. PSPNet (Zhao et al., 2017), on the other hand, is used for scene analysis of semantic segmentation. It employs a pyramid pooling module that connects four global pooling layers of different sizes in parallel and pools the original feature map to generate different levels. These levels are then restored to their original size after convolution and upsampling. DeepLabv2 (Chen et al., 2017), a semantic segmentation model, proposes an atrous spatial convolution pooling module (ASPP) inspired by SPPNet. ASPP uses multiple parallel atrous convolutional layers with different sampling rates and convolution kernels with different receptive fields. The features extracted for each sampling rate are further processed in separate branches and fused to generate the final result. Finally, Feature Pyramid is a high-level feature representation method commonly used in computer vision, proposed by Lin et al. (2017a). It enhances the model’s ability to process features of different scales by pyramidizing low-level features into high-level feature representations. The methods discussed earlier can partially mitigate the problem of excessive size variation in detected objects. The FPN adopts a top-down approach to process feature maps, merging both high and low-level features. This strategy maintains robust semantic content while optimizing both processing speed and resource efficiency. Given considerations like memory allocation, model intricacy, and prediction duration, the FPN emerges as an apt selection for this study.

Attention mechanism

The attention mechanism in computer vision can be seen as a dynamic selection process that adaptively weights features based on their importance. To accomplish this goal, the spatial transformer network (STN) (Jaderberg et al., 2015) developed by Google DeepMind plays a crucial role. This network is designed to convert spatial information from the initial image into an alternative space. By doing so, it effectively captures the deformations within the input while preserving essential information. Another approach, SENET (Hu, Shen & Sun, 2018), establishes interdependencies between feature channels through squeeze and excitation operations. By doing so, the network can automatically learn the importance of each channel and enhance useful features while suppressing less useful ones for the current task. The Convolutional Block Attention Module (CBAM) (Woo et al., 2018) expands the channel attention module and spatial attention module to process the input feature layer. Huang et al. (2019) proposed the Criss-Cross Attention, which efficiently models adaptive full-image context by decomposing the core self-attention structure in Transformer into row-wise self-attention and column-wise self-attention. This method approximates a dense self-attention calculation using two consecutive sparse self-attention calculations. Long-distance dependencies can capture contextual information beneficial for surgical instrument segmentation tasks. However, methods based on attention mechanisms require the generation of large attention maps to compute the relationships between each pixel, leading to high computational complexity. Criss-Cross Attention calculates the relationship between pixels in the rows and columns where each pixel resides, significantly reducing computational complexity while obtaining semantic information. This meets the requirement of network lightweighting in this article and enhances the network’s focus on surgical instruments.

Proposed methods

Overall architecture

In the following sections, we present a detailed description of our proposed method for surgical instrument segmentation, which is both lightweight and efficient. The overall framework is illustrated in Fig. 1. Our network is an end-to-end encoder-decoder architecture. For the encoder part, we adopt the LDCNet encoder proposed in a previous work as the backbone network to extract features. The backbone network employs convolutions of different sizes to group and capture features of varying sizes, thereby reducing the number of parameters and improving computational efficiency. We provide further details on this in the next subsection.

Figure 1 Overall architecture of the proposed method.

Image source credit: Mengyu, Xiaoxiang, Zhoujin, Yitong, & Liping (https://doi.org/10.5281/zenodo.8322390). CC-BY 4.0. https://creativecommons.org/licenses/by/4.0/legalcode.

After the feature extraction process of LDCNet, we obtain four levels of feature maps. These feature maps are then fed into the corresponding 1 × 1 convolution operation of the FPN. This network performs multi-scale operations to improve the model’s ability to detect surgical instruments of varying shapes and sizes. The output upsampling modules of the four levels are combined to create a single feature map with dimensions of 64 × 64 × 64. Inter-channel splicing is applied to this feature map. The upsampling module consists of 3 × 3 depthwise separable convolutions, 1 × 1 convolutions, and 2 × upsampling using bilinear interpolation. The spliced feature map is then passed through the cross attention module, which allows the decoder to better capture information from the encoder and enhance the feature representation capability. Finally, the prediction map is generated by applying 3 × 3 depth separable convolution, 1 × 1 convolution, and upsampling operations.

In the entire model structure, depth separable convolution is employed to decrease network parameters and computations, while enhancing the model’s generalization ability. The separable convolution block divides the conventional convolution operation into two steps: depthwise convolution and pointwise convolution. This approach constitutes a more efficient convolution method.

The LDCNet encoder design is inspired by DenseNet, ResNeXt and ConvNeXt (Liu et al., 2022). It has been further improved upon to enhance its performance. The overall framework of the encoder is shown in Fig. 2. DenseNet’s unique mechanism of dense connection enables effective utilization of features from different levels. This leads to a smoother decision boundary and robust performance, even when training data is limited. On the other hand, ResNeXt introduces the concept of parallel convolution by combining outputs from multiple small convolution kernels. This enhances the network’s ability to capture information and refine edges. The LDCNet encoder inherits the advantages of DenseNet, such as alleviating gradient disappearance, preserving low-dimensional features, and reducing parameters. It also incorporates the concept of ConvNeXt, resulting in further improvement in model performance.

Figure 2 The structure of LDCNet.

Image source credit: Mengyu, Xiaoxiang, Zhoujin, Yitong, & Liping (https://doi.org/10.5281/zenodo.8322390). CC-BY 4.0. https://creativecommons.org/licenses/by/4.0/legalcode.

LDCNet

ConvNeXt is a convolutional network that combines the successful approaches of Vision Transformer and CNN models. It has demonstrated superior performance compared to complex Transformer-based models, highlighting the effectiveness of optimizing the original CNN technology and parameters to achieve state-of-the-art results. In the macro design of ConvNeXt, the stacking ratio of multi-level blocks is 1:1:3:1. The third stage contains a larger number of stacked blocks, which enhances the model’s accuracy. Similarly, in the design of LDCNet, we adopted a comparable stacking ratio of blocks in each stage, with specific layer numbers set as 2, 2, 6, and 2, respectively. Figure 3A illustrates that ConvNeXt utilizes 3 × 3 depthwise separable convolution to create an Inverted bottleneck block. This design effectively increases the model’s depth and width while avoiding information loss caused by compression. The inverted block employs a larger convolution kernel in the middle to enhance the depth of the feature map, while smaller convolution kernels are used at both ends to preserve the resolution of the feature map. This design enables the feature map to be more adaptable when converting between feature spaces of different dimensions, ultimately leading to improved model performance. Drawing inspiration from the multi-scale feature extraction capability of the Inception block in GoogLeNet (Szegedy et al., 2015), we propose a new bottleneck layer with two branches. One branch utilizes a depthwise separable convolution with a 7 × 7 kernel, while the other branch employs a depthwise separable convolution with a 3 × 3 kernel. The output feature maps of these two branches are summed and concatenated with the input feature maps to generate the output feature maps of the proposed bottleneck layer, as depicted in Fig. 3B.

Figure 3 The structure of BottleNeck.

Criss-cross attention module

Dense contextual information, derived from the long-distance dependence between features, plays a crucial role in enhancing the model’s understanding of images. In surgical operations, the shape and size of instruments undergo constant changes, and the instrument tip remains in motion. Therefore, it becomes crucial to improve the capability of the decoder-side feature map. This improvement is aimed at capturing information effectively from the encoder side. To address this, this article introduces a CCAM in the decoder side, which is a self-attention mechanism. The introduced module is from CCNet, whitch incorporates both row attention and column attention to obtain global information. Moreover, the network parameters are minimized to meet the lightweight requirements of this study.

The specific calculation process of CCAM is shown in Fig. 4. By applying three convolutionals with 1 × 1 filters to the input feature map F∈RC×W×H, three feature maps, Q, K and V, are obtained. Where {Q,K}∈RB×W×H and both C and B represent feature channel numbers (B < C). At the position index u on the W, H plane, vectors from the feature map K that align with the row and column of u contribute to the vector set Ωu∈R(H+W−1)×B. Following this, the vector Q∈RB is accquired. The attention matrix du is then derived via the Affinity operation, as defined by:

Figure 4 The structure of CCAM.

(1) du=ΩuQu

where du∈RH+W−1 reflects the degree of correlation between Qu and ΩuUpon applying the Softmax operation to du, the attention weight matrix Au∈RH+W−1 for the location u is obtained. At the same time, through another 1 × 1 convolution on F, the feature map V∈RC×W×H is produced. Vectors from the corresponding position u in V that intersect row and column are gathered to form the vector set Φu∈R(H+W−1)×C. Finally, by performing the Aggregation operation with Au, the complete output feature map Fu′ is generated. The equation is as follows:

(2) Fu′=ΦuTAu+Fu

Our model can achieve high-precision edge segmentation of surgical instruments, mainly due to the effective combination of the lightweight backbone network LDCNet, FPN, and CCAM. LDCNet is improved based on DenseNet, using two branch bottleneck layers to replace the ordinary convolutional layers in DenseBlock. This reduces the model parameters, increases the receptive field of the model, and enhances the model’s ability to recognize targets of different sizes. The FPN module integrates four different levels of feature maps output by the LDCNet encoder. This effectively integrates context information of different scales, helping to improve the model’s ability to recognize surgical instruments in complex scenarios. The CCAM can enhance the network’s perception of important spatial position information and improve the model’s segmentation performance. In our model, the feature map stitched by the FPN module is used as input to the CCAM module. The CCAM module enhances the model’s perception of important position information by learning attention weights, further improving the segmentation accuracy of the model. Additionally, we use depthwise separable convolutions throughout the model structure. This is done to reduce the computational complexity and the number of parameters of the model, making our model an efficient lightweight network. These design choices enable our model to achieve high-precision edge segmentation of surgical instruments, while also considering the computational efficiency of the model.

Experiment

Datasets

Kvasir-Instrument: The Kvasir-Instrument dataset was obtained by Jha et al. (2021). through endoscopic examinations conducted at Bærum Hospital in Norway. The dataset comprises images and videos captured using standard endoscopic equipment used for gastrointestinal surgery. All data used in this study were collected in compliance with the Bærum Hospital patient informed consent protocol. The dataset includes 590 frames, each annotated with various GI surgical tools corresponding to the original image. The image resolution ranges from 720 × 576 to 1,280 × 1,024. To ensure the dataset’s accuracy and reliability, two professional research assistants flagged the real data, which was subsequently reviewed and corrected by a gastroenterologist.

EndoVis2017: The Endovis2017 dataset (Allan et al., 2019) is a component of the Robotic Instrument Segmentation sub-challenge, which was part of the Endoscopic Vision Challenge in 2017. The dataset comprises 10 series of pig abdominal surgeries that were recorded using the da Vinci system. These surgeries employed a variety of surgical tools, including Large Needle Driver, Prograsp Forceps, Monopolar Curved Scissors, Cadiere Forceps, and Bipolar Forceps. The images were annotated by a team of experts trained by Intuitive Surgical. The dataset includes the first eight series of original endoscopic images and annotated images, totaling 1,800 surgical images with a resolution of 1280 × 1204 pixels.

Private dataset: The Private dataset was collected from the digestive endoscopic surgical robot platform developed under the National Key R&D Program project (2018YFB1307700). Pig stomachs were used to simulate human tissue, and Meilan reagent was injected to make small pieces of tissue swell to simulate lesion bodies. The team operated the surgical robot to remove the lesion bodies in vitro by extending the robotic arm into the pig stomach. A total of 2,344 frames were selected from the endoscopic recording video, each containing two types of surgical instruments: electric knives and electric clips. The image resolution is 1,312 × 1,020. Using the labelme annotation tool, 2,344 images were annotated under the guidance of professional doctors to ensure accuracy.

To facilitate a more robust comparison of model performance, we harmonized the image resolution across the three datasets to 256 × 256 prior to model integration. This specification optimally balances GPU memory consumption and segmentation efficacy. Using established methodologies, we subdivided the dataset into training, validation, and test subsets. Initially, the dataset underwent a randomized shuffle to safeguard its randomness and mitigate potential biases from data sequencing. A random number generator was employed for this purpose, thereby ensuring consistent replicability in experimental outcomes. Subsequently, the reorganized dataset was portioned as follows: 20% designated for validation to fine-tune model parameters and facilitate selection, 20% reserved for testing to gauge the model’s conclusive performance, and the residual 60% allocated for training purposes.

Training details

Our proposed model was implemented using the PyTorch 1.12.0 deep learning framework, Python 3.8, and the PyTorch Lightning 1.7.7 framework, which offers efficient and convenient features for deep learning research. The model was trained on a computer equipped with an AMD R5-5600 6-core processor, an 8 GB Nvidia RTX 3060Ti GPU, and 16 GB of RAM. We trained the model on different datasets for 100 epochs, with validation occurring every two epochs and the best weight parameters saved five times. The resulting five sets of weight parameters were then used to predict the test set, and the resulting indicators were averaged. To optimize memory usage, we established distinct parameters for different network models. Specifically, we utilized 16 threads for the data reading program. The initial learning rate was set at 1e-3, and the learning rate was dynamically adjusted through the ReduceLROnPlateau strategy. We utilized the AdamW (Loshchilov & Hutter, 2017) optimizer and set the training period at 100. Additionally, we utilized automatic mixed precision training and the FocalLoss (Lin et al., 2017b) loss function, which reduces the weight of easily classified samples while increasing the weight of difficult-to-classify samples. The FocalLoss function is expressed as follows:

(3) FL(pt)=−αt(1−pt)γlog(pt)

p∈[0,1] represents the estimated probability of the model for a labeled class. The parameter γ is an adjustable focusing parameter, primarily designed to minimize the impact of easily classified samples on the overall loss. In the original FocalLoss article, the authors demonstrated through experiments that when γ is set to 2, the model can achieve better results. α is a balancing parameter, used to adjust the weight between positive and negative samples. During our experiments, we found that the combination of setting y to 2 and α to 0.25 yields favorable segmentation outcomes.

Evaluation metrics

To evaluate the performance of the proposed model in all aspects, we introduced commonly used evaluation metrics in semantic segmentation to compare with other existing mainstream methods. Based on the confusion matrix, we used four metrics- precision, recall, F1-score, and mIoU to examine the accuracy of the model’s predictions from multiple perspectives. TP represents the number of true positive classes, TN represents the number of true negative classes, FP represents the number of false positive classes, and FN represents the number of false negative classes. In the context of surgical instrument image segmentation discussed in this article, TP represents the pixels correctly predicted as belonging to the surgical instrument, TN denotes the pixels correctly predicted as part of the background tissue, FP signifies the pixels inaccurately predicted as surgical instrument, and FN indicates the pixels erroneously predicted as background tissue.

Precision represents the proportion of true positive samples among the samples recognized as positive by the model. In general, the higher the precision, the better the performance of the model.

(4) Precision=TPTP+FP

Recall represents the ratio of the number of positive samples correctly identified by the model to the total number of positive samples. In general, the higher the recall, the better the performance of the model, as it indicates that more positive samples are predicted correctly by the model.

(5) Recall=TPTP+FN

F1-score is defined as the harmonic mean of precision and recall, and its value ranges from 0 to 1, with 1 being the best and 0 being the worst.

(6) F1=2Precision×RecallPrecision+Recall=2TP2TP+FP+FN

The mIoU is a widely used benchmark across various datasets, and is commonly used as the primary evaluation metric for image semantic segmentation models in academic articles. mIoU represents the average IoU ratio of the predicted and true masks for each class in the dataset. With k being the number of classes, mIoU provides a valuable measure of segmentation accuracy.

(7) mIoU=1k+1∑i=0kTPTP+FN+FP

Results on the private dataset

In this section, we conduct a comprehensive evaluation of our proposed lightweight semantic segmentation network, comparing it with several other popular models on a private dataset. Specifically, we compare our method with UNet (Ronneberger, Fischer & Brox, 2015), TransUNet (Chen et al., 2021), CGNet (Wu et al., 2020), SegFormer (Xie et al., 2021), GCNet (Cao et al., 2019), ContextNet (Poudel et al., 2018), DABNet (Li et al., 2019), and FPENet (Liu & Yin, 2019), where the backbones of UNet and GCNet use ResNet18. Our proposed network outperforms all other models on the private dataset we constructed, achieving an mIoU of 97.11%, F1-score of 98.52%, precision of 98.59%, and recall of 98.46%, as shown in Table 1. ContextNet performs the second best, with a slight decrease in mIoU, F1-score, precision, and recall compared to our method.

Table 1 Quantitative results on the private dataset (numbers in bold are the best indicator results).

Method	mIoU (%)	Mean F1 (%)	Mean precision (%)	Mean recall (%)	
UNet	96.45	98.18	98.11	98.25	
TransUNet	93.72	96.70	97.10	96.35	
CGNet	95.57	97.71	97.54	97.87	
SegFormer	95.11	97.46	97.31	97.61	
GCNet	95.23	97.52	97.89	97.17	
ContextNet	96.57	98.24	98.28	98.20	
DABNet	96.32	98.11	97.86	98.36	
FPENet	96.42	98.16	98.22	98.10	
Ours	97.11	98.52	98.59	98.46	

In our presentation of the qualitative analysis, three representative images from each dataset were chosen. These images, detailed in Figs. 5–7, are organized into three rows. The initial row displays the overlaid segmentation outcomes from different models juxtaposed with the original images, highlighting the precise positioning of the surgical instruments and the relative efficacy of each segmentation model. The subsequent row emphasizes specific highlighted details. By amplifying these areas, we elucidate the model’s segmentation finesse in intricate regions. Such insights are instrumental in evaluating the model’s proficiency in interpreting intricate elements like edges, shapes, and textures. The final row delineates the unaltered segmentation results from assorted models. Specifically, Fig. 6, derived from a private dataset, suggests our method’s superior handling of instrument edges in low-light conditions, achieving a comprehensive contour without an excessive number of false-positive predictions.

Figure 5 Qualitative results on the private dataset.

Image source credit: Mengyu Zhou, Xiaoxiang Han, Zhoujin Liu, Yitong Chen, & Liping Sun (https://doi.org/10.5281/zenodo.8098618). CC-BY 4.0. https://creativecommons.org/licenses/by/4.0/legalcode.

Figure 6 Qualitative results on the public dataset MICCAI EndoVis2017.

Image source credit: Mengyu, Xiaoxiang, Zhoujin, Yitong, & Liping (https://doi.org/10.5281/zenodo.8322280). CC-BY 4.0. https://creativecommons.org/licenses/by/4.0/legalcode.

Figure 7 Qualitative results on the public dataset Kvasir-instrument.

Image source credit: Mengyu, Xiaoxiang, Zhoujin, Yitong, & Liping (https://doi.org/10.5281/zenodo.8322390). CC-BY 4.0. https://creativecommons.org/licenses/by/4.0/legalcode.

Results on the public dataset MICCAI EndoVis2017

Our proposed method was evaluated on the public dataset EndoVis2017, using the same comparison models as mentioned earlier. The results are presented in Table 2. In terms of all metrics, our method outperforms all other models. It achieves an mIoU of 95.83%, which is 0.31% higher than the second-place ContextNet. Additionally, it achieves an F1-score of 97.84%, which is 0.25% higher than the second-place DABNet, a precision of 97.69%, which is 0.18% higher than the second-place DABNet, and a recall of 98.0%. These results demonstrate the strong performance of our method on the public dataset EndoVis2017. Figure 6 provides a visual comparison of the models on the EndoVis2017 dataset. Within the enlarged section, our model adeptly identifies the shape and position of the instrument’s tip, closely mirroring the provided annotations and manifesting exceptional accuracy. Precise delineation of the instrument tip is paramount in surgical instrument image segmentation. This tip often represents the point of interaction between the instrument and tissue, and its accurate representation is vital for surgical precision and safety. While the tip’s diminutive size and irregular contour challenge many existing methods, leading to imprecise segmentation, our model effectively discerns these intricate details, underscoring the robustness of our approach in this domain.

Table 2 Quantitative results on the public dataset MICCAI EndoVis2017 (numbers in bold are the best indicator results).

Method	mIoU (%)	Mean F1 (%)	Mean Precision (%)	Mean Recall (%)	
UNet	95.11	97.45	97.34	97.57	
TransUNet	94.83	97.30	97.34	97.26	
CGNet	93.42	96.52	96.51	96.53	
SegFormer	93.85	96.76	97.70	96.82	
GCNet	94.25	96.98	96.19	97.81	
ContextNet	94.52	97.13	97.18	97.08	
DABNet	95.37	97.59	97.51	97.68	
FPENet	94.41	97.07	97.05	97.09	
Ours	95.83	97.84	97.69	98.00	

Results on the public dataset Kvasir-instrument

The Kvasir-instrument dataset contains fewer images compared to the previous two datasets, with only 471 images used for training. As a result, the overall data accuracy is lower. The experimental comparison results are shown in Table 3. Our proposed method outperforms the second-place DABNet method in terms of various metrics. Specifically, our method achieves a slightly higher mIoU (0.02%), F1-score (0.02%), and precision (0.1%), while the recall is slightly lower (0.05%). Although the difference with the second-place method is not significant on this dataset, our method accurately segments surgical instruments. Figure 7 presents the qualitative outcomes derived from the Kvasir-instrument dataset. A detailed analysis of these results reveals that our technique effectively segments slender surgical instruments without causing discontinuities or making unwarranted predictions. Such efficiency is anchored in our model’s notable resistance to noise and its adeptness in handling intricate details throughout both the feature extraction and prediction processes. A comparative assessment with other methodologies, as evident from the Fig. 7, underscores that our method delineates edge information with enhanced precision, yielding more distinct, fluid, and accurate segmentation contours. This proficiency largely stems from our model’s ability to amalgamate features across varying scales and its rigorous focus on local details.

Table 3 Quantitative results on the public dataset Kvasir-instrument (numbers in bold are the best indicator results).

Method	mIoU (%)	Mean F1 (%)	Mean precision (%)	Mean recall (%)	
UNet	91.91	95.66	95.93	95.40	
TransUNet	91.60	95.48	96.08	94.90	
CGNet	92.59	96.05	96.76	95.38	
SegFormer	91.57	95.46	96.08	94.87	
GCNet	92.14	95.79	96.67	94.96	
ContextNet	91.84	95.62	96.44	94.84	
DABNet	93.27	96.43	96.91	95.95	
FPENet	92.05	95.75	96.36	95.15	
Ours	93.29	96.45	97.01	95.90	

In this study, we report significant outcomes across three distinct datasets: two public and one proprietary. Spanning a range of surgical scenarios and instruments, these datasets approximate real-world surgical data. Our results underscore the model’s robust adaptability to varied data distributions and features. The model incorporates the LDCNet encoder with two bottleneck layer branches: one with a 7 ×7 and the other with a 3 × 3 depthwise separable convolution. This architecture facilitates the extraction of multi-scale features, adeptly capturing both nuanced local details and overarching global context, enhancing its compatibility with diverse dataset images. Furthermore, the use of depthwise separable convolution efficiently trims the parameter count, thereby reducing computational demands. This streamlining not only optimizes the model for scalability across datasets of different magnitudes but also bolsters its generalizability. Overall, our model demonstrates notable flexibility, seamlessly integrating with a variety of dataset images.

Model efficiency evaluation

In surgical assistance systems, the appraisal of model performance transcends mere accuracy and identification capabilities, encompassing its real-world operational efficiency. Critical performance metrics include the model’s parameter count, floating point operations (FLOPs), and frames per second (FPS). The model’s size, gauged by its parameter count, serves as a pivotal metric for evaluating its complexity and memory demands. FLOPs quantify the model’s computational demands, shedding light on its processing efficiency, while FPS benchmarks its execution speed-crucial for real-time applications. Efficiency metrics for several models are presented in Table 4. Notably, our proposed model ranks second with 466K parameters, a 0.202 FLOPs score, and a 280.417 FPS count. While the FPENet model exhibits superior efficiency, a review of Tables 1–3 underscores our model’s superiority in terms of accuracy. Weighing both efficiency and performance, our model stands out as the prime selection. It adeptly segments surgical instruments and facilitates seamless deployment in engineering settings. In essence, the model masterfully balances precision and efficiency, presenting a robust solution for surgical assistance systems.

Table 4 Comparison of model efficiency (data in bold are ranked first, those underlined are ranked second).

Method	Parameters	FLOPs	FPS	
UNet	31 M	16.415	161.325	
TransUNet	67.1 M	32.506	89.840	
CGNet	491 K	0.854	152.966	
SegFormer	7.7 M	3.272	174.886	
GCNnet	61.4M	2.425	237.387	
ContextNet	911 K	0.212	318.380	
DABNet	752 K	1.283	263.743	
FPENet	114 K	0.183	184.508	
Ours	466 K	0.202	280.417	

Results and analysis of ablation experiments

To enhance the accuracy of surgical instrument segmentation while meeting the deployment requirements of the surgical assistance system, we have proposed a lightweight semantic segmentation network. We utilized LDCNet as the backbone network and integrated FPN and CCAM modules to improve the model’s performance. To analyze the contribution of each network block to the overall model, we conducted ablation experiments to evaluate the performance of each component.

In establishing our baseline model, we omitted the FPN and CCAM components from the proposed approach to more effectively ascertain their individual influences. The ablation study results, presented in Table 5, revealed that the integration of the FPN module into the baseline model led to an mIoU enhancement of 0.03% and 0.05% across two public datasets. This enhancement is attributable to the FPN module’s proficiency in managing multi-scale features. By channeling feature maps from diverse LDCNet layers into the FPN, we facilitated feature fusion and upsampling, yielding a more comprehensive multi-scale feature representation. Incorporating the CCAM module into the proposed model, we observed a 0.12% increase in mIoU relative to the baseline model, as evidenced by experimental trials conducted on the Endovis2017 dataset. A t-test comparing the experimental outcomes of the two models revealed a statistically significant difference (p <0.05). This module excels in capturing contextual details and dynamically modulating feature channel weights, thus bolstering the model’s segmentation prowess.

Table 5 Results of ablation experiments (numbers in bold are the best indicator results).

Dataset	Method	Backbone	mIoU(%)	mF1(%)	mPrecision(%)	mRecall(%)	
Kvasir-instrument	Baseline	LDCNet	93.22	95.90	96.76	95.08	
Baseline with FPN	LDCNet	93.27	96.41	96.99	96.04	
Baseline with CCAM	LDCNet	92.83	96.19	96.56	95.83	
Ours	DenseNet	90.85	90.05	95.37	94.74	
Ours	ResNet	92.94	96.25	96.60	95.92	
Ours	LDCNet	93.29	96.45	97.01	95.90	
Endovis2017	Baseline	LDCNet	95.63	97.73	97.58	97.89	
Baseline with FPN	LDCNet	95.66	97.62	97.52	97.98	
Baseline with CCAM	LDCNet	95.75	97.80	97.50	98.10	
Ours	DenseNet	95.00	97.39	97.22	97.57	
Ours	ResNet	95.54	97.68	97.74	97.89	
Ours	LDCNet	95.85	97.84	97.69	98.00	

Further, we evaluated the influence of various backbone networks on model efficacy. As delineated in Table 5, the LDCNet backbone outperformed both resnet and densenet on the aforementioned datasets. This superior performance stems from LDCNet’s refined architecture, derived from DenseNet, which incorporates dual bottleneck layer branches in lieu of conventional convolutional layers present in DenseBlock. This adaptation augments the model’s capability to discern features across multiple scales. To conclude, our innovative method, underscored by judicious design and synergistic module interaction, has markedly elevated the segmentation performance for surgical instruments. The ablation study robustly corroborates the pivotal role of each module in this performance enhancement, underscoring our model’s stature as a streamlined yet potent network.

Discussion

Our research methodology showcased superior accuracy in surgical instrument segmentation across three distinct datasets. It not only outperformed other models in processing edge details but also benefited from a more streamlined architecture, with fewer parameters and enhanced efficiency. Such attributes are particularly beneficial for integration into surgical assistance systems, facilitating the automation of surgical robots and fortifying surgical safety.

While our model outshined in accuracy across the three datasets and secured a commendable second place in efficiency, our study is not without limitations. Notably, our model exhibits pronounced advantages with larger datasets. This could be attributed to its intricate design: when presented with expansive datasets, it can harness a wealth of data to discern intricate features and patterns, which, in turn, bolsters its performance. In contrast, with smaller datasets, there is a risk of the model overfitting, compromising its ability to adapt to novel data. The larger datasets, given their potential for more balanced data representation across categories, can counteract the challenges of class imbalance, facilitating the model’s capacity to discern features across various categories, hence refining segmentation accuracy. Furthermore, despite our model’s exemplary performance on the selected datasets, they may not wholly capture the multifaceted nature of genuine surgical scenarios. To navigate the challenges of overfitting and augment real-world adaptability, strategies such as data augmentation, the incorporation of regularization techniques, or expanded data training might be explored. Pursuing the acquisition of more authentic surgical scenario data remains pivotal to further substantiate the model’s viability and efficacy.

In subsequent research, we aim to optimize our model’s structure further and investigate lighter backbone networks. We also plan to devise methods for the efficient extraction of robust semantic features and integrate transfer learning to bolster the model’s generalization capacity. Our methodology is scalable to diverse medical imaging tasks, providing an avenue to evaluate its generalizability. Such an approach is pivotal for applications in real-time surgical navigation and the analysis of surgical videos. To conclude, while our method has showcased commendable performance in the segmentation of surgical instruments, it is not devoid of challenges and limitations. We anticipate that future research will furnish a comprehensive evaluation of our technique, addressing existing limitations.

Conclusions

In this study, we present a lightweight yet effective network for surgical instrument segmentation. Our network incorporates self-attention mechanisms to automate surgical instrument localization. To reduce model parameters and extract features, we utilize the high-performance backbone network LDCNet. The network effectively integrates context and multi-scale information through the FPN module, addressing the issue of sudden changes in surgical instrument size during endoscopy. In order to enhance the capture of detailed features, we introduce the CCAM module prior to outputting the results, enabling the network to prioritize surgical instruments. Experimental results demonstrate that our proposed method surpasses several popular models on a private dataset, achieving remarkable accuracy with an mIoU of 97.11%, F1-score of 98.52%, precision of 98.59%, and recall of 98.46%. Furthermore, our method performs competitively on public datasets. We are confident that this work establishes a strong foundation for future research in surgical instrument segmentation.

Supplemental Information

Supplemental Information 1 Source code.

Click here for additional data file.

Additional Information and Declarations

Competing Interests

Author Contributions

Data Availability

The authors declare that they have no competing interests.

Mengyu Zhou conceived and designed the experiments, performed the experiments, performed the computation work, prepared figures and/or tables, authored or reviewed drafts of the article, and approved the final draft.

Xiaoxiang Han conceived and designed the experiments, performed the experiments, performed the computation work, prepared figures and/or tables, authored or reviewed drafts of the article, and approved the final draft.

Zhoujin Liu analyzed the data, prepared figures and/or tables, and approved the final draft.

Yitong Chen analyzed the data, prepared figures and/or tables, and approved the final draft.

Liping Sun conceived and designed the experiments, authored or reviewed drafts of the article, and approved the final draft.

The following information was supplied regarding data availability:

The raw data are available at Zenodo:

Figure 5: Mengyu Zhou, Xiaoxiang Han, Zhoujin Liu, Yitong Chen, & Liping Sun. (2023). The surgical instrument dataset in the article A Lightweight Segmentation Network for Endoscopic Surgical Instruments Based on Edge Refinement and Efficient Self-Attention [Data set]. Zenodo. https://doi.org/10.5281/zenodo.8098618. CC-BY4.0. https://creativecommons.org/licenses/by/4.0/legalcode.

Figure 6: Mengyu, Xiaoxiang, Zhoujin, Yitong, & Liping. (2023). Our processed Kvasir-Instrument dataset in the article A Lightweight Segmentation Network for Endoscopic Surgical Instruments Based on Edge Refinement and Efficient Self-Attention [Data set]. Zenodo. https://doi.org/10.5281/zenodo.8322390. CC-BY4.0. https://creativecommons.org/licenses/by/4.0/legalcode.

Figures 1, 2, 7: Mengyu, Xiaoxiang, Zhoujin, Yitong, & Liping. (2023). Our processed EndoVis2017 dataset in the article A Lightweight Segmentation Network for Endoscopic Surgical Instruments Based on Edge Refinement and Efficient Self-Attention [Data set]. Zenodo. https://doi.org/10.5281/zenodo.8322280. CC-BY4.0. https://creativecommons.org/licenses/by/4.0/legalcode.

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
