# Peer review of "A lightweight segmentation network for endoscopic surgical instruments based on edge refinement and efficient self-attention"

_PeerJ Computer Science, doi:10.7717/peerj-cs.1746_

## Round 0.1 · original submission · Major Revisions

The reviewers have substantial concerns about this manuscript. The authors should provide point-to-point responses to address all the concerns and provide a revised manuscript with the revised parts being marked in different color.

**Language Note:** PeerJ staff have identified that the English language needs to be improved. When you prepare your next revision, please either (i) have a colleague who is proficient in English and familiar with the subject matter review your manuscript, or (ii) contact a professional editing service to review your manuscript. PeerJ can provide language editing services - you can contact us at copyediting@peerj.com for pricing (be sure to provide your manuscript number and title). – PeerJ Staff

Reviewer 1 ·

Basic reporting

It is truly remarkable to witness the application of edge refinement and efficient self-attention based methods in surgical instrument segmentation. The achievement of a remarkable 97.11% mean Intersection over Union (mIoU) with just 469K parameters is highly impressive.

However, to ensure the completeness of the content, there are a few essential points that need to be addressed.

Experimental design

In line 139, the introduction of the proposed methods, overall architecture, LDCNet, and Criss-cross attention module should be combined with the presentation of experimental data to provide a more comprehensive understanding of the research.

Regarding line 274, it would be beneficial to clarify the specific rationale behind setting

Validity of the findings

Regarding Figure 5, it is recommended to add an additional second column to the figure that includes overlaid masks on the original images. This would provide a more visual representation of the segmentation results and further enhance the clarity and impact of the figure.

Lastly, it would be beneficial to include a section on future studies or future directions before the conclusion, particularly since the model's application focuses on robot-assisted surgical systems. This section could outline potential areas of further investigation, potential improvements, or future applications of the proposed method.

By addressing these points, we can improve the clarity and completeness of the content without exceeding the word count.

Reviewer 2 ·

Basic reporting

The overall paper is well-written but still has room for improvement. The introduction of the Criss-Cross Attention Module and LDCNet in the context of surgical instrument segmentation seems a novel and well-thought-out approach. The overall structure appears to be clear and organized. Simpler terminology could help make the paper accessible to a broader audience. The discussion sections could benefit from a more comprehensive evaluation of the results and future research directions. potential limitations should be discussed more openly. Limitations such as overfitting should be carefully addressed.

Experimental design

1. you have to modify the purpose of why you want to do segmentation from lines 46 to 54. The motivation for the development of the segmentation tool is not clear.
2. For lines 98-139, the suitability for surgical instrument segmentation should be discussed for the three different types of models separately.
3. Can you describe and justify your stratification method for separating the test set and training set?
4. you have to define how you defined TN, TP FP, and FN in the context of surgical instrument segmentation.
5. need to work on the legends of the images. e.g. specify what the abbreviation of "GT" stands for

Validity of the findings

6. What is the adaptivity of this model with the new dataset?
7. Can you provide empirical evidence by demonstrating the performance in real-world scenarios?
8. should provide a thorough discussion of the specific design choices and their impact on the results for the lightweight backbone network, LDCnet, and the cooperation between FPN and CCAM modules.
9. The discussion lacks a forward-looking perspective on potential limitations and areas for improvement. It is more important to understand the challenge and limitations of this technique.

Reviewer 3 ·

Basic reporting

The article is generally well written, the proposed method is clearly written and experimental methods are nicely put together with ablation studies to support the network decision choices.

There are a number of minor issues:
- through multi-scale and attention mechanism, the network performs better in segmentation details. But in the article, it claims edge refinement. I would expect some explicit edge refinement process
- Because lightweight usually goes side by side with inference speed, author should include the FLOPs and FPS in the inference scenes
- Although I understand the intuition based on my understand of self attention, Criss-cross attention part need to be reframed and better explained. Figure 4 could also be improved, attention A and V interaction is not represented in the figure.
- there are a number typos in the article, ex: line 220 w*h and 222 W*H should be the same; Table 5 has 'beseline' and 'LDCnet'

Experimental design

As mentioned above, the experiments are very well put together
- relatively detailed training strategies are deployed and most of the design choices are good and well thought out
- would like to see why Focal Loss instead of Soft Dice loss
- IoU and F1 are good metrics for evaluating segmentation results

On ther other hand, experiments should include the FLOPs and FPS in the inference for better comparisons of lightweight networks.

Validity of the findings

The work itself is an incremental improvement towards more efficient segmentation network. References to two public datasets used in the work, as well as the source code is available along side the article.

Annotated reviews are not available for download in order to protect the identity of reviewers who chose to remain anonymous.

---

## Round 0.2 · Minor Revisions

There are remaining minor revisions that need to be addressed. The authors should provide point-to-point responses to address all the concerns and provide a revised manuscript with the revised parts being marked in different color.

Reviewer 1 ·

Basic reporting

After carefully reviewing the article, I found that all the comments have been addressed. The paper is now ready for publication.

Experimental design

N/A

Validity of the findings

N/A

Reviewer 2 ·

Basic reporting

The paper focused on a lightweight segmentation network for endoscopic surgical instruments with similar performance as other studies. Several techniques were blended in this study. However, the use of certain methodologies, like the application of LDCNet initially for remote sensing, requires clearer justification, especially in terms of the advantages claimed in the introduction. Despite these minor issues, the paper overall presents solid research, and with slight revisions, it is deemed fit for publication.

Experimental design

1. The paper mentions that the proposed method outperforms existing models. Please justify how you define 'outperformed' with an improvement of 0.12% MIoU. Perhaps some citations or a well-designed statistical analysis may help.

Validity of the findings

1. You need a citation for the claim ‘The current methods are not detailed enough for the segmentation of surgical instruments at the edges.
2. You have to explain your rationale about why edge detection is important in segmentation in your introduction.
3. You have claimed that the surgical instruments may appear white and blurred in your introduction. Can you demonstrate an example of such type or artifacts and showcase the improvement with your algorithm?
4. Connections between the technologies mentioned (e.g., da Vinci surgical robot, LDCNet) and the main problem of the paper should be made more explicit.

---

## Round 0.3 · accepted · Accept

Reviewers are satisfied with the revisions, and I concur to recommend accepting this manuscript.

Reviewer 3 ·

Basic reporting

The revised version have addressed all the issued mentioned in the comments.

Specifically, I see the motivation is now better and addressing more on the efficiency of the network as well as performance, which makes the paper much better motivated. There is also a better and more detailed explanation of the network modules and better illustrative figures.

I also see that the FLOPs and speed of the networks addressed in the results section.

The paper is ready of publishing.

Experimental design

na

Validity of the findings

na